# Information Geometry of Orthogonal Initializations and Training

**Piotr Aleksander Sokół and Il Memming Park**
Department of Neurobiology and Behavior
Departments of Applied Mathematics and Statistics, and Electrical and Computer Engineering
Institutes for Advanced Computing Science and AI-driven Discovery and Innovation
Stony Brook University, Stony Brook, NY 11733
`{memming.park, piotr.sokol}@stonybrook.edu`

## Abstract

Recently mean field theory has been successfully used to analyze properties of wide, random neural networks. It gave rise to a prescriptive theory for initializing feed-forward neural networks with orthogonal weights, which ensures that both the forward propagated activations and the backpropagated gradients are near $\ell_2$ isometries and as a consequence training is orders of magnitude faster. Despite strong empirical performance, the mechanisms by which critical initializations confer an advantage in the optimization of deep neural networks are poorly understood. Here we show a novel connection between the maximum curvature of the optimization landscape (gradient smoothness) as measured by the Fisher information matrix (FIM) and the spectral radius of the input-output Jacobian, which partially explains why more isometric networks can train much faster. Furthermore, given that orthogonal weights are necessary to ensure that gradient norms are approximately preserved at initialization, we experimentally investigate the benefits of maintaining orthogonality throughout training, and we conclude that manifold optimization of weights performs well regardless of the smoothness of the gradients. Moreover, we observe a surprising yet robust behavior of highly isometric initializations — even though such networks have a lower FIM condition number *at initialization*, and therefore by analogy to convex functions should be easier to optimize, experimentally they prove to be much harder to train with stochastic gradient descent. We conjecture the FIM condition number plays a non-trivial role in the optimization.

## 1 Introduction

Deep neural networks (DNN) have shown tremendous success in computer vision problems, speech recognition, amortized probabilistic inference, and the modelling of neural data. Despite their performance, DNNs face obstacles in their practical application, which stem from both the excessive computational cost of running gradient descent for a large number of epochs, as well as the inherent brittleness of gradient descent applied to very deep models. A number of heuristic approaches such as batch normalization, weight normalization and residual connections (He et al., 2016; Ioffe & Szegedy, 2015; Salimans & Kingma, 2016) have emerged in an attempt to address these trainability issues. Recently mean field theory has been successful in developing a more principled analysis of gradients of neural networks, and has become the basis for a new random initialization principle. The mean field approach postulates that in the limit of infinitely wide random weight matrices, the distribution of pre-activations converges weakly to a Gaussian. Using this approach, a series of works proposed to initialize the networks in such a way that for each layer the input-output Jacobian has mean singular values of 1 (Schoenholz et al., 2017). This requirement was further strengthened to suggest that the spectrum of singular values of the input-output Jacobian should concentrate on 1, and it was shown that this can only be achieved with random orthogonal weight matrices.

Under these conditions the backpropagated gradients are bounded in $\ell_2$ norm (Pennington et al., 2017) irrespective of depth, i.e., they neither vanish nor explode. It was shown experimentally in (Pennington et al., 2017; Xiao et al., 2018; Chen et al., 2018) that networks with these *critical* initial conditions train orders of magnitude faster than networks with arbitrary initializations. The empirical success invites questions from an optimization perspective on how the spectrum of the hidden layer

input-output Jacobian relates to notions of curvature of the parameters space, and consequentially to convergence rate. The largest effective (initial) step size $\eta_0$ for stochastic gradient descent is inversely proportional to the local gradient smoothness $M$ (Bottou et al., 2018; Boyd & Vandenberghe, 2004). Intuitively, the gradient step can be at most as large as the fastest change in the parameter landscape. Recent attempts have been made to analyze the mean field geometry of the optimization using the Fisher information matrix (FIM) (Amari et al., 2019; Karakida et al., 2019). The theoretical and practical appeal of measuring curvature with the FIM is due to among other reasons the fact that the FIM is necessarily positive (semi-)definite even for non-convex objectives, and due to it its intimate relationship with the Hessian matrix. Karakida et al. (2019) derived an upper bound on the maximum eigenvalue, however this bound is not satisfactory since it is agnostic of the entire spectrum of singular values and therefore cannot differentiate between Gaussian and orthogonal weight initalizations.

In this paper, we develop a new bound on the parameter curvature $M$ given the maximum eigenvalue of the Fisher information $\lambda_{\max}(\mathbf{G})$ which holds for random neural networks with both Gaussian and orthogonal weights. We derive this quantity to inspect the relation between the singular value distribution of the input-output Jacobian and locally maximal curvature of the parameter space. We use this result to probe different orthogonal, nearly isometric initializations, and observe that, broadly speaking, networks with a smaller initial curvature train faster and generalize better, as expected. However, consistent with a previous report (Pennington et al., 2018), we also observe highly isometric networks perform worse despite having a slowly varying loss landscape ( i.e. small initial $\lambda_{\max}(\mathbf{G})$). We conjecture that the long term optimization behavior is depends on trivially on the smallest eigenvalue $m$ and therefore surprisingly there is a *sweetspot with the condition number being* $\frac{m}{M} > 1$.

We then investigate whether constraining the spectrum of the Jacobian matrix of each layer affects optimization rate. We do so by training networks using Riemannian optimization to constrain their weights to be orthogonal, or nearly orthogonal and we find that manifold constrained networks are insensitive to the maximal curvature at the beginning of training unlike the unconstrained gradient descent (hereafter "Euclidean"). In particular, we observe that the advantage conferred by optimizing over manifolds cannot be explained by the improvement of the gradient smoothness as measured by $\lambda_{\max}(\mathbf{G})$. Finally, we observe that contrary to Bansal et al. (2018)'s results Euclidean networks with a carefully designed initialization reduce the test misclassification error at approximately the same rate as their manifold constrained counterparts, and overall attain a higher accuracy.

## 2 BACKGROUND

### 2.1 FORMAL DESCRIPTION OF THE NETWORK

Following (Pennington et al., 2017; 2018; Schoenholz et al., 2017), we consider a feed-forward, fully connected neural network with $L$ hidden layers. Each layer $l \in \{1, \ldots, L\}$ is given as a recursion of the form

$$\mathbf{x}^l = \phi(\mathbf{h}^l), \quad \mathbf{h}^l = \mathbf{W}^l \mathbf{x}^{l-1} + \mathbf{b}^l \tag{1}$$

where $\mathbf{x}^l$ are the activations, $\mathbf{h}^l$ are the pre-activations, $\mathbf{W}^l \in \mathbb{R}^{N^l \times N^{l-1}}$ are the weight matrices, $\mathbf{b}^l$ are the bias vectors, and $\phi(\cdot)$ is the activation function. The input is denoted as $\mathbf{x}^0$. The output layer of the network computes $\hat{\mathbf{y}} = g^{-1}(\mathbf{h}^g)$ where $g$ is the link function of some generalized linear model (GLM) and $\mathbf{h}^g = \mathbf{W}^g x^L + \mathbf{b}^g$.

The hidden layer input-output Jacobian matrix $\mathbf{J}_{\mathbf{x}^0}^{\mathbf{x}^L}$ is,

$$\mathbf{J}_{\mathbf{x}^0}^{\mathbf{x}^L} \triangleq \frac{\partial \mathbf{x}^L}{\partial \mathbf{x}^0} = \prod_{l=1}^{L} \mathbf{D}^l \mathbf{W}^l \tag{2}$$

where $\mathbf{D}^l$ is a diagonal matrix with entries $\mathbf{D}_{i,i}^l = \phi'(\mathbf{h}_i^l)$. As pointed out in (Pennington et al., 2017; Schoenholz et al., 2017), the conditioning of the Jacobian matrix affects the conditioning of the back-propagated gradients for all layers.

### 2.2 CRITICAL INITIALIZATIONS

Extending the classic result on the Gaussian process limit for wide layer width obtained by (Neal, 1996), recent work (Matthews et al., 2018; Lee et al., 2018) has shown that for deep untrained networks with elements of their weight matrices $\mathbf{W}_{i,j}$ drawn from a Gaussian distribution $\mathcal{N}(0, \frac{\sigma_\mathbf{W}^2}{N^l})$

the empirical distribution of the pre-activations $\mathbf{h}^l$ converges weakly to a Gaussian distribution $\mathcal{N}(0, q^l \mathbf{I})$ for each layer $l$ in the limit of the width $N \to \infty$. Similarly, it has been postulated that random orthogonal matrices scaled by $\sigma_{\mathbf{W}}$ give rise to the same limit. Under this mean-field condition, the variance of the pre-activation distribution $q^l$ is recursively given by,

$$q^l = \sigma_{\mathbf{W}}^2 \int \phi\left(\sqrt{q^{l-1}}h\right) \mathrm{d}\mu(h) + \sigma_{\mathbf{b}}^2 \tag{3}$$

where $\mu(h)$ denotes the standard Gaussian measure $\int \frac{\mathrm{d}h}{\sqrt{2\pi}} \exp\left(\frac{-h^2}{2}\right)$ and $\sigma_{\mathbf{b}}^2$ denotes the variance of the Gaussian distributed biases (Schoenholz et al., 2017). The variance of the first layer pre-activations $q^1$ depends on $\ell_2$ norm squared of inputs $q^1 = \frac{\sigma_{\mathbf{W}}^2}{N^1} \|p(\mathbf{x}^0)\|_2^2 + \sigma_{\mathbf{b}}^2$. The recursion defined in equation 3 has a fixed point

$$q^* = \sigma_{\mathbf{W}}^2 \int \phi\left(\sqrt{q^*}h\right) \mathrm{d}\mu(h) + \sigma_{\mathbf{b}}^2 \tag{4}$$

which can be satisfied for all layers by appropriately choosing $\sigma_{\mathbf{W}}, \sigma_{\mathbf{b}}$ and scaling the input $\mathbf{x}^0$. To permit the mean field analysis of backpropagated signals, the authors (Schoenholz et al., 2017; Pennington et al., 2017; 2018; Karakida et al., 2019) further assume the propagated activations and back propagated gradients to be independent. Specifically,

**Assumption 1.** *[Mean field assumptions]*
*(i)* $\lim_{N \to \infty} \mathbf{h} \xrightarrow{d} \mathcal{N}(0, q^*)$
*(ii)* $\lim_{N \to \infty} \mathrm{Cov}\left[\mathbf{J}_{\mathbf{x}^{i+1}}^g \mathbf{h}^i, \mathbf{J}_{\mathbf{x}^{j+1}}^g \mathbf{h}^j\right] = 0$   *for all $i \neq j$*

Under this assumption, the authors (Schoenholz et al., 2017; Pennington et al., 2017) analyze distributions of singular values of Jacobian matrices between different layers in terms of a small number of parameters, with the calculations of the backpropagated signals proceeding in a selfsame fashion as calculations for the forward propagation of activations. The corollaries of Assumption 1 and condition in equation 4 is that $\phi'(\mathbf{h}^l)$ for $1 \leq l \leq L$ are i.i.d. In order to ensure that $\mathbf{J}_{\mathbf{x}^0}^{\mathbf{x}^L}$ is well conditioned, (Pennington et al., 2017) require that in addition to the variance of pre-activation being constant for all layers, two additional constraints be met. Firstly, they require that the mean square singular value of $\mathbf{DW}$ for each layer has a certain value in expectation.

$$\chi = \frac{1}{N} \mathbb{E}\left[\mathrm{Tr}\left[(\mathbf{DW})^\top \mathbf{DW}\right]\right] = \sigma_{\mathbf{W}}^2 \int \left[\phi'(\sqrt{q^*}h)\right]^2 \mathrm{d}\mu(h) \tag{5}$$

Given that the mean squared singular value of the Jacobian matrix $\mathbf{J}_{\mathbf{x}^0}^{\mathbf{x}^L}$ is $(\chi)^L$, setting $\chi = 1$ corresponds to a critical initialization where the gradients are asymptotically stable as $L \to \infty$. Secondly, they require that the maximal squared singular value $s_{\max}^2$ of the Jacobian $\mathbf{J}_{\mathbf{x}^0}^{\mathbf{x}^L}$ be bounded. Pennington et al. (2017) showed that for weights with Gaussian distributed elements, the maximal singular value increases linearly in depth even if the network is initialized with $\chi = 1$. Fortunately, for orthogonal weights, the maximal singular value $s_{\max}$ is bounded even as $L \to \infty$ (Pennington et al., 2018).

## 3 THEORETICAL RESULTS: RELATING THE SPECTRA OF JACOBIAN AND FISHER INFORMATION MATRICES

To better understand the geometry of the optimization landscape, we wish to put a Lipschitz bound on the gradient, which in turn gives an upper bound on the largest step size of any first order optimization algorithm. For a general objective function $f$, the condition is equivalent to

$$\|\nabla f(x) - \nabla f(x')\|_2 \leq M \|x - x'\|_2 \quad \text{for all} \quad x, x' \subset \mathcal{S} \subseteq \mathbb{R}^d$$

The Lipschitz constant ensures that the gradient doesn't change arbitrarily fast with respect to $x$, $x'$, and therefore $\nabla f$ defines a descent direction for the objective over a distance $M$. In general estimating the Lipschitz constant is NP-hard (Kunstner et al., 2019), therefore we seek to find local measures of curvature along the optimization trajectory. As we will show below the approximate gradient smoothness is tractable for randomly initialized neural networks. The analytical study of Hessians of random neural networks started with (Pennington & Bahri, 2017), but was limited to shallow architectures. Subsequent work by Amari et al. (2019) and Karakida et al. (2019) on

second order geometry of random networks shares much of the spirit of the current work, in that it proposes to replace the possibly indefinite Hessian with the related Fisher information matrix as a measure of curvature. The Fisher information matrix plays a fundamental role in the geometry of probabilistic models under the Kullback-Leibler divergence loss — it defines a (local) Riemannian metric, which in turn defines distances on the manifold of probability distributions generated by the model. Notably, the FIM does not define a unique metric on this statistical manifold, and alternative notions of intrinsic curvature can be derived by replacing the Kullback-Leibler divergence with the 2-Wasserstein distance (Li & Montúfar, 2018). Moreover, since the Fisher information matrix bears a special relation to the Hessian it can also be seen as defining an approximate curvature matrix for second order optimization. Recall that the FIM is defined as

**Definition.** *Fisher Information Matrix*

$$\mathbf{G} \triangleq \mathbb{E}_{p_\theta(\mathbf{y}|\mathbf{x}^0)} \left[ \mathbb{E}_{p(\mathbf{x}^0)} \left[ \nabla_\theta \log p_\theta(\mathbf{y}|\mathbf{x}^0) \nabla_\theta \log p_\theta(\mathbf{y}|\mathbf{x}^0)^\top \right] \right] \tag{6}$$

$$= \mathbb{E}_{p_\theta(\mathbf{y}|\mathbf{x}^0)} \left[ \mathbb{E}_{p(\mathbf{x}^0)} \left[ \mathbf{J}_\theta^{h^g\top} \nabla_{h^g}^2 \mathcal{L} \, \mathbf{J}_\theta^{h^g} \right] \right] = \mathbb{E}_{p_\theta(\mathbf{y}|\mathbf{x}^0)} \left[ \mathbb{E}_{p(\mathbf{x}^0)} \left[ \mathbf{H} - \sum_k \nabla_{\mathbf{x}^g} \mathcal{L}_k \, \nabla_\theta^2 h_k^g \right] \right] \tag{7}$$

where $\mathcal{L}$ denotes the loss and $\mathbf{h}^g$ is the output layer. The relation between the Hessian and Fisher Information matrices is apparent from equation 7, showing that the Hessian $\mathbf{H}$ is a quadratic form of the Jacobian matrices plus the possibly indefinite matrix of second derivatives with respect to parameters.

Our goal is to express the gradient smoothness using the results of the previous section. Given equation 7 we can derive an analytical approximation to the Lipschitz bound using the results from the previous section; i.e. we will express the expected maximum eigenvalue of the random Fisher information matrix in terms of the expected maximum singular value of the Jacobian $\mathbf{J}_{\mathbf{h}^1}^{\mathbf{h}^L}$. To do so, let us consider the output of a multilayer perceptron as defining a conditional probability distribution $p_\theta(\mathbf{y}|\mathbf{x}^0)$, where $\Theta = \{\text{vec}(\mathbf{W}^1), \ldots, \text{vec}(\mathbf{W}^L), \mathbf{b}^1, \ldots, \mathbf{b}^L\}$ is the set of all hidden layer parameters, and $\theta$ is the column vector containing the concatenation of all the parameters in $\Theta$. As observed by Martens & Grosse (2015) the Fisher of a multilayer network naturally has a block structure, with each corresponding to the weights and biases of each layer. These blocks with respect to parameter vectors $a, b \in \Theta$ can further be expressed as

$$\mathbf{G}_{a,b} = \mathbf{J}_a^{\mathbf{h}^g\top} \mathbf{H}_g \mathbf{J}_b^{\mathbf{h}^g} \tag{8}$$

where the final layer Hessian $\mathbf{H}_g$ is defined as $\nabla_{\mathbf{h}^g}^2 \log p_\theta(\mathbf{y}|\mathbf{x}^0)$. We can re-express the outer product of the score function $\nabla_{\mathbf{h}^g} \log p_\theta(\mathbf{y}|\mathbf{x}^0)$ as the second derivative of the log-likelihood (see equation 6), provided it satisfies certain technical conditions. What is important for us is that all canonical link function for generalized linear models, like the softmax function and the identity function allow this re-writing, and that this re-writing allows us drop the conditional expectation with respect to $p_\theta(\mathbf{y}|\mathbf{x}^0)$. The Jacobians in equation 8 can be computed iteratively. Importantly the Jacobian from the output layer to the $a$-th parameter block is just the product of diagonal activations and weight matrices multiplied by the Jacobian from the $\alpha$-th layer to the $a$-th parameter. We define these matrices of partial derivatives of the $\alpha$-th layer pre-activations with respect to the layer specific parameters separately for $\mathbf{W}^\alpha$ and $\mathbf{b}^\alpha$ as:

$$\mathbf{J}_a^{\mathbf{h}^\alpha} = \mathbf{x}^{\alpha-1\top} \otimes \mathbf{I} \qquad\qquad \text{for } a = \text{vec}(\mathbf{W}^\alpha) \tag{9}$$

$$\mathbf{J}_a^{\mathbf{h}^\alpha} = \mathbf{I} \qquad\qquad \text{for } a = \mathbf{b}^\alpha \tag{10}$$

Under the infinitesmally weak correlation assumption (see Assumption 1), we can further simplify the expression for the blocks of the Fisher information matrix equation 8.

**Lemma 1.** *The expected blocks with respect to weight matrices for all layers $\alpha, \beta \neq 1$ are*

$$\mathbf{G}_{\text{vec}(\mathbf{W}^\alpha), \text{vec}(\mathbf{W}^\beta)} = \mathbb{E}\left[ \mathbf{x}^{\alpha-1} \mathbf{x}^{\beta-1\top} \right] \otimes \mathbb{E}\left[ \mathbf{J}_{\mathbf{h}^\alpha}^{\mathbf{h}^g\top} \mathbf{H}_g \mathbf{J}_{\mathbf{h}^\beta}^{\mathbf{h}} \right] \tag{11}$$

**Lemma 2.** *The expected blocks with respect to a weight matrix $\mathbf{W}^\alpha$ and a bias vector $\mathbf{b}^\beta$ are*

$$\mathbf{G}_{\text{vec}(\mathbf{W}^\alpha), \mathbf{b}^\beta} = \mathbb{E}\left[ \mathbf{x}^{\alpha-1\top} \otimes \mathbf{I} \right] \mathbb{E}\left[ \mathbf{J}_{\mathbf{h}^\alpha}^{\mathbf{h}^g\top} \mathbf{H}_g \mathbf{J}_{\mathbf{h}^\beta}^{\mathbf{h}^g} \right] \tag{12}$$

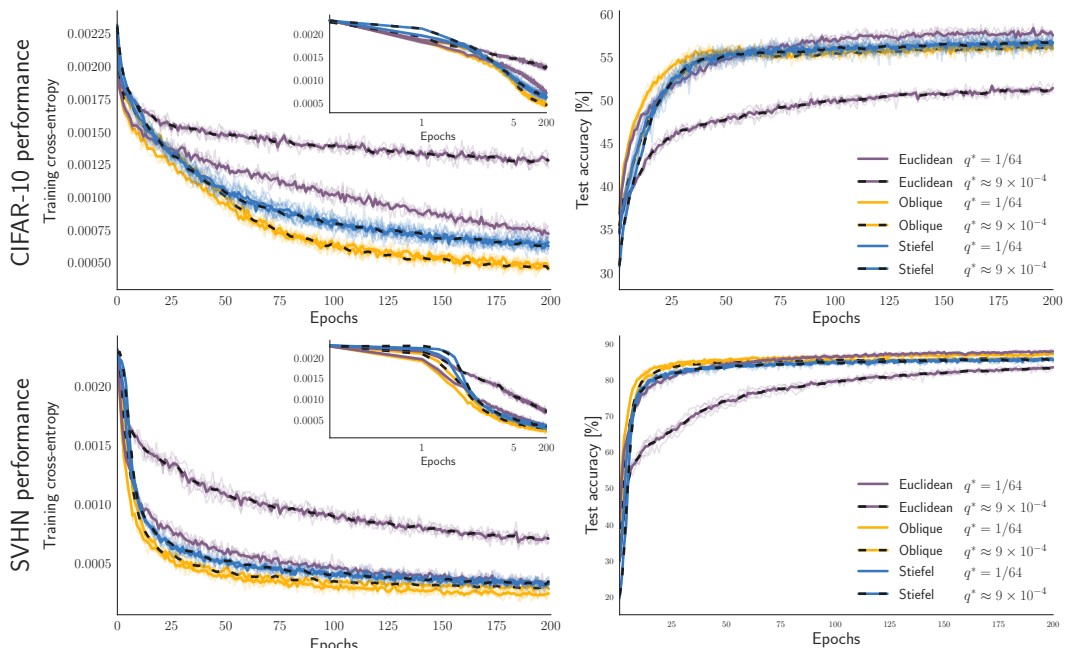

Figure 1: **Network with manifold constrained weights are relatively insensitive to the choice of initial weight scaling** $q^*$: We compare training loss and test accuracy for Euclidean, Stiefel and Oblique networks with two different values of $q^*$. The manifold constrained networks minimize the training loss at approximately the same rate, being faster than both Euclidean networks. Despite this, there is little difference between the test accuracy of the Stiefel and Oblique networks and the Euclidean networks initialized with $q^* = 1/64$. Notably, the latter attains a marginally higher test set accuracy towards the end of training.

The crucial observation here is that in the mean-field limit the expectation of the product of activations $\mathbf{x}^{\alpha-1}$, $\mathbf{x}^{\beta-1}$ is either zero or rank 1 for activations in different layers. The case when both activations are in the same layer is trivially taken care of by our mean-field assumptions — the term is equal to the 2nd non-central moment, i.e. the covariance plus potentially a rank one mean term.

Now, leveraging Lemmas 1 and 2 we derive a block diagonal approximation which in turn allows us to bound the maximum eigenvalue $\lambda_{\max}(\mathbf{G})$. In doing so we will use a corollary of the block Gershgorin theorem.

**Proposition 1** ((informal) Block Gershgorin theorem). *The maximum eigenvalue $\lambda_{\max}(\mathbf{G})$ is contained in a union of disks centered around the maximal eigenvalue of each diagonal block with radia equal to the sum of the singular values of the off-diagonal terms.*

For a rigorous statement of the theorem see Appendix A.1. It is noteworthy that block-diagonal approximations have been crucial to the application of Fisher Information matrices as preconditioners in stochastic second order methods (Botev et al., 2017; Martens & Grosse, 2015). These methods were motivated by practical performance in their choice of number of diagonal blocks used for preconditioning. Under the mean-field assumptions we are able to show computable bounds on the error in approximating the spectrum of the Fisher information matrix. The proposition 1 suggest a simple, easily computable way to bound the expected maximal eigenvalue of the Fisher information matrix—choose the block with the largest eigenvalue and expected spectral radia for the corresponding off diagonal terms. We do so by making an auxiliary assumption:

**Assumption 2.** *The maximum singular value of $\mathbf{J}_{\mathbf{h}^\alpha}^{\mathbf{h}^g}$ monotonically increases as $\alpha \downarrow 1$.*

We motivate this assumption in a twofold fashion: firstly the work done by Pennington et al. (2017; 2018) shows that the spectral edge, i.e. the maximal, non-negative singular value in the support of of the spectral distribution increases with depth, secondly it has been commonly observed in numerical experiments that very deep neural networks have ill conditioned gradients.

Under this assumption it is sufficient to study the maximal singular value of blocks of the Fisher information matrix with respect to $\text{vec}(\mathbf{W}^1), b^1$ and the spectral norms of its corresponding off-diagonal blocks. We define functions $\Sigma_{\max}$ of each block as upper bounds on the spectral bounds of the respective block. The specific values are given in the following Lemma:

**Lemma 3.** *The maximum expected singular values of the off-diagonal blocks $\forall \beta \neq 1$ are bounded by $\Sigma_{\max}(\cdot)$ :*
*for weight-to-weight blocks*

$$\mathbb{E}\left[\sigma_{\max}\left(\mathbf{G}_{\text{vec}(\mathbf{W}^1),\text{vec}(\mathbf{W}^\beta)}\right)\right] \leq \Sigma_{\max}\left(\mathbf{G}_{\text{vec}(\mathbf{W}^1),\text{vec}(\mathbf{W}^\beta)}\right) \tag{13}$$

$$\triangleq \sqrt{N^\beta}\left|\mathbb{E}\left[\phi(h)\right]\right|\left\|\mathbb{E}\left[\mathbf{x}^0\right]\right\|_2 \mathbb{E}\left[\sigma_{\max}\left(\mathbf{J}_{\mathbf{h}^1}^{\mathbf{h}^g\top}\right)\right]\mathbb{E}\left[\sigma_{\max}\left(\mathbf{H}_g\right)\right]\mathbb{E}\left[\sigma_{\max}\left(\mathbf{J}_{\mathbf{h}^\beta}^{\mathbf{h}^g}\right)\right] \tag{14}$$

*for weight-to-bias blocks*

$$\mathbb{E}\left[\sigma_{\max}\left(\mathbf{G}_{\text{vec}(\mathbf{W}^1),b^\beta}\right)\right] \leq \Sigma_{\max}\left(\mathbf{G}_{\text{vec}(\mathbf{W}^1),b^\beta}\right) \tag{15}$$

$$\triangleq \left|\mathbb{E}\left[\phi(h)\right]\right|\mathbb{E}\left[\sigma_{\max}\left(\mathbf{J}_{\mathbf{h}^1}^{\mathbf{h}^g\top}\right)\right]\mathbb{E}\left[\sigma_{\max}\left(\mathbf{H}_g\right)\right]\mathbb{E}\left[\sigma_{\max}\left(\mathbf{J}_{\mathbf{h}^\beta}^{\mathbf{h}^g}\right)\right] \tag{16}$$

*and for bias-to-bias blocks*

$$\mathbb{E}\left[\sigma_{\max}\left(\mathbf{G}_{b^1,b^\beta}\right)\right] \leq \Sigma_{\max}\left(\mathbf{G}_{b^1,b^\beta}\right) \triangleq \mathbb{E}\left[\sigma_{\max}\left(\mathbf{J}_{\mathbf{h}^1}^{\mathbf{h}^g\top}\right)\right]\mathbb{E}\left[\sigma_{\max}\left(\mathbf{H}_g\right)\right]\mathbb{E}\left[\sigma_{\max}\left(\mathbf{J}_{\mathbf{h}^\beta}^{\mathbf{h}^g}\right)\right] \tag{17}$$

*For proof see Appendix A.2*

Note that the expectations for layers $> 1$ is over random networks realizations and averaged over data $\mathbf{x}^0$; i.e. they are taken with respect to the Gaussian measure, whereas the expectation for first layer weights is taken with respect to the empirical distribution of $\mathbf{x}^0$ (see equation 4). Depending on the choice of $q^*$ and therefore implicitly both the rescaling of $\mathbf{x}^0$ and the values of $\mathbb{E}[\phi(\mathbf{h})]$ the singular values of the weight blocks might dominate those associated with biases dominate — compare equation 14 and equation 17.

**Theorem** (Bound on the Fisher Information Eigenvalues). *If $\left\|\mathbb{E}\left[\mathbf{x}^0\right]\right\|_2 \leq 1$ then eigenvalue associated with $b^1$ will dominate, giving an upper bound on $\lambda_{\max}(\mathbf{G})$*

$$\mathbb{E}\left[\lambda_{\max}(\mathbf{G})\right] \leq \mathbb{E}\left[\sigma_{\max}\left(\mathbf{G}_{b^1,b^1}\right)\right] + \Sigma_{\max}\left(\mathbf{G}_{b^1,\text{vec}(\mathbf{W}^1)}\right)$$
$$+ \sum_{\beta>1}\Sigma_{\max}\left(\mathbf{G}_{b^1,b^\beta}\right) + \Sigma_{\max}\left(\mathbf{G}_{\text{vec}(b^1),\text{vec}(\mathbf{W}^\beta)}\right)$$

*otherwise the maximal eigenvalue of the FIM is bounded by*

$$\mathbb{E}\left[\lambda_{\max}(\mathbf{G})\right] \leq \mathbb{E}\left[\sigma_{\max}\left(\mathbf{G}_{\text{vec}(\mathbf{W}^1),\text{vec}(\mathbf{W}^1)}\right)\right] + \Sigma_{\max}\left(\mathbf{G}_{b^1,\text{vec}(\mathbf{W}^1)}\right) + \sum_{\beta>1}\Sigma_{\max}\left(\mathbf{G}_{\text{vec}(\mathbf{W}^1),b^\beta}\right)$$
$$+ \Sigma_{\max}\left(\mathbf{G}_{\text{vec}(\mathbf{W}^1),\text{vec}(\mathbf{W}^\beta)}\right)$$

*Moreover, it is interesting to note two things. Firstly, $\mathbb{E}\left[\sigma_{\max}\left(\mathbf{J}_{\mathbf{h}^1}^{\mathbf{h}^g}\right)\right]$ factor appear in all the above summands. Secondly, we can bound $\sigma_{\max}$ for the diagonal blocks with $\mathbb{E}\left[\lambda_{\max}\left(\mathbf{H}_g\right)\right]\mathbb{E}\left[\sigma_{\max}\left(\mathbf{J}_{\mathbf{h}^1}^{\mathbf{h}^g}\right)\right]^2$. These two fact reveal that the FIM maximum eigenvalue is upper upperbounded by a quadratic function of the spectral radius of the input-output Jacobian; i.e. $\lambda_{\max}(\mathbf{G})$ is $\mathcal{O}\left(\sigma_{\max}^2(\mathbf{J}_{\mathbf{h}^1}^{\mathbf{h}^g})\right)$.*
*For proof see Appendix A.4*

The functional form of the bound is essentially quadratic in $\mathbb{E}\left[\sigma_{\max}(\mathbf{J}_{\mathbf{h}^1}^{\mathbf{h}^g})\right]$ since the term appears in the summand as with powers at most two. This result shows that the strong smoothness, given by the maximum eigenvalue of the FIM, is *proportional to* the squared maximum singular value of the input-output Jacobian (see Fig. 2). Moreover, the bound essentially depends on $q^*$ via the expectation $\mathbb{E}[\phi(h)]$, through $\mathbf{J}_{\mathbf{h}^1}^{\mathbf{h}^g}$ and implicitly through $\mathbf{H}_g$. For regression problems this dependence is monotonically increasing in $q^*$ (Pennington et al., 2018; 2017) since $\mathbf{H}_g$ is just the identity. However, this does not hold for all generalized linear models since $\lambda_{\max}(\mathbf{H}_g)$ is not necessarily a monotonically increasing function of the pre-activation variance at layer $\mathbf{h}^g$. We demonstrate this in the case of softmax regression in the Appendix A.3. Finally, to obtain a specific bound on $\lambda_{\max}(\mathbf{G})$ we might consider bounding each $\mathbb{E}\left[\sigma_{\max}(\mathbf{J}_{\mathbf{h}^\alpha}^{\mathbf{h}^g})\right]$ appearing in Theorem 3 in terms of its Frobenius norm. The corresponding result is the eigenvalue bound derived by Karakida et al. (2019).

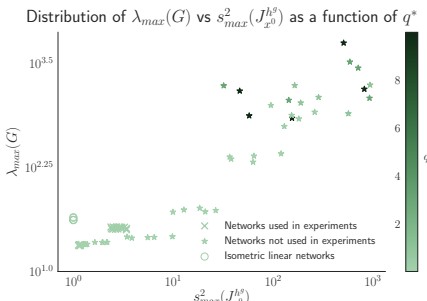 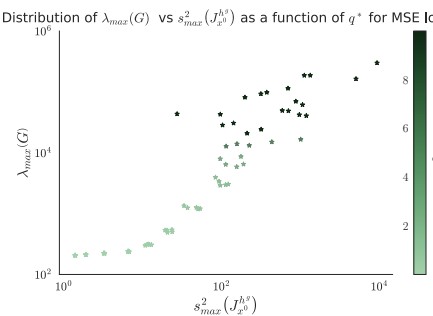

Figure 2: **Left:** at initialization the maximum curvature of the loss landscape (measured by the $\lambda_{\max}$ of the Fisher correlates highly ($\rho = 0.65$) with the maximum squared singular value of the Jacobian $\mathbf{J}^{\mathbf{h}^g}_{\mathbf{x}^0}$. The choice of choice of the preactivation variance, $q^*$ affects not only the conditioning of the gradients but also the gradient Lipschitz constant. **Right:** we verify this separately for mean-squared error, for which a strictly monotonic relationship between $q^*$ and $\lambda_{\max}$ and $\mathbf{J}^{\mathbf{h}^g}_{\mathbf{x}^0}$ is predicted, given that the Hessian of the output layer is the identity. This is corroborated by a correlation coefficient of $\rho = 0.81$.

### 3.1 NUMERICAL EXPERIMENTS

We first verify the bound derived in section 3 for networks with random orthogonal weights. We then numerically investigate the behavior of the maximum FIM eigenvalue during training with particular attention being paid to the possible benefits of maintaining orthogonality or near orthogonality during optimization in relation to unconstrained networks. Following Pennington et al. (2017) we trained a 200 layer $\tanh$ network on CIFAR-10 and SVHN[1] and we set the width of each layer to be $N = 400$ and chose the $\sigma_{\mathbf{W}}$, $\sigma_{\mathbf{b}}$ in such a way to ensure that mean singular value of the input-output Jacobian concentrates on 1 but $s^2_{\max}$ varies as a function of $q^*$ (see Fig. 2). We considered four different critical initializations with $q^* = \left[10^{-4}, \frac{1}{64}, \frac{1}{2}, 8\right]$, which differ both in spread of the singular values as well as in the resulting training speed and final test accuracy as reported by (Pennington et al., 2017). In the main text we focus on the smaller values since those networks should be closer to being isometric and therefore, by our theory, ought to train better. The remaining two networks with $q^* = \left[\frac{1}{2}, 8\right]$ are presented in the Appendix A.5. To test how enforcing strict orthogonality or near orthogonality affects convergence speed and the maximum eigenvalues of the Fisher information matrix, we trained Stiefel and Oblique constrained networks and compared them to the unconstrained "Euclidean" network described in (Pennington et al., 2017). We used a Riemannian version of ADAM (Kingma & Ba, 2015). When performing gradient descent on non-Euclidean manifolds, we split the variables into three groups: (1) Euclidean variables (e.g. the weights of the classifier layer, biases), (2) non-negative scaling $\sigma_{\mathbf{W}}$ both optimized using the regular version of ADAM, and (3) manifold variables optimized using Riemannian ADAM. The initial learning rates for all the groups, as well as the non-orthogonality penalty (see 43) for Oblique networks were chosen via Bayesian optimization, maximizing validation set accuracy after 50 epochs. All networks were trained with a minibatch size of 1000. We trained 5 networks of each kind, and collected eigenvalue and singular value statistics every 5 epochs, from the first to the fiftieth, and then after the hundredth and two hundredth epochs.

Based on the bound on the maximum eigenvalue of the Fisher information matrix derived in Section 3, we predicted that at initialization $\lambda_{\max}(\mathbf{G})$ should covary with $\sigma^2_{\max}(\mathbf{J}^{\mathbf{h}^g}_{\mathbf{x}^0})$. We tested our prediction using the empirical Fisher information matrix (Kunstner et al., 2019) and we find a significant correlation between the two (Pearson coefficient $\rho = 0.64$). We additionally test the relation for mean-squared error loss, and observe that our theoretical predictions hold better in this case — the discrepancy can be attributed to the effect of the Hessian of the output layer, and its non-monotonic relation between preactivation variance and maximum eigenvalue of the Hessian of the GLM layer. The numerical values are presented in Fig. 2. Additionally we see that both the maximum singular value and maximum eigenvalue increase monotonically as a function of $q^*$. Motivated by the previous work by (Saxe et al., 2014) showing depth independent learning dynamics in linear orthogonal networks, we included 5 instantiations of this model in the comparison. The input to the linear

---

[1]Code available on: `https://github.com/PiotrSokol/info-geom`

network was normalized the same way as the critical, non-linear networks with $q^* = 1/64$. The deep linear networks had a substantially larger $\lambda_{\max}(\mathbf{G})$ than its non-linear counterparts initialized with identically scaled input (Fig. 2). Having established a connection between $q^*$ the maximum singular value of the hidden layer input-output Jacobian and the maximum eigenvalue of the Fisher information, we investigate the effects of initialization on subsequent optimization. As reported by (Pennington et al., 2017), the learning speed and generalization peak at intermediate values of $q^* \approx 10^{-0.5}$. This result is counter intuitive given that the maximum eigenvalue of the Fisher information matrix, much like that of the Hessian in convex optimization, upper bounds the maximal learning rate (Boyd & Vandenberghe, 2004; Bottou et al., 2018). To gain insight into the effects of the choice of $q^*$ on the convergence rate, we trained the Euclidean networks and estimated the local values of $\lambda_{\max}$ during optimization. At the same time we asked whether we can effectively control the two aforesaid quantities by constraining the weights of each layer to be orthogonal or near orthogonal. To this end we trained Stiefel and Oblique networks and recorded the same statistics. We present training results in Fig. 1, where it can be seen that Euclidean networks with $q^* \approx 9 \times 10^{-4}$ perform worse with respect to training loss and test accuracy than those initialized with $q^* = 1/64$. On the other hand, manifold constrained networks are insensitive to the choice of $q^*$. Moreover, Stiefel and Oblique networks perform marginally worse on the test set compared to the Euclidean network with $q^* = 1/64$, despite attaining a lower training loss. We observe that reduced performance of Euclidean networks initialized with $q^* \approx 9 \times 10^{-4}$ may partially be explained by their rapid increase in $\lambda_{\max}(\mathbf{G})$ within the initial 5 epochs of optimization (see Fig. 3 in the Appendix). While all networks undergo this rapid increase, it is most pronounced for Euclidean networks with $q^* \approx 9 \times 10^{-4}$. The increase $\lambda_{\max}(\mathbf{G})$ correlates with the inflection point in the training loss curve that can be seen in the inset of Fig. 1. Interestingly, the manifold constrained networks optimize efficiently despite differences in $\lambda_{max}(\mathbf{G})$ (Kohler et al., 2019).

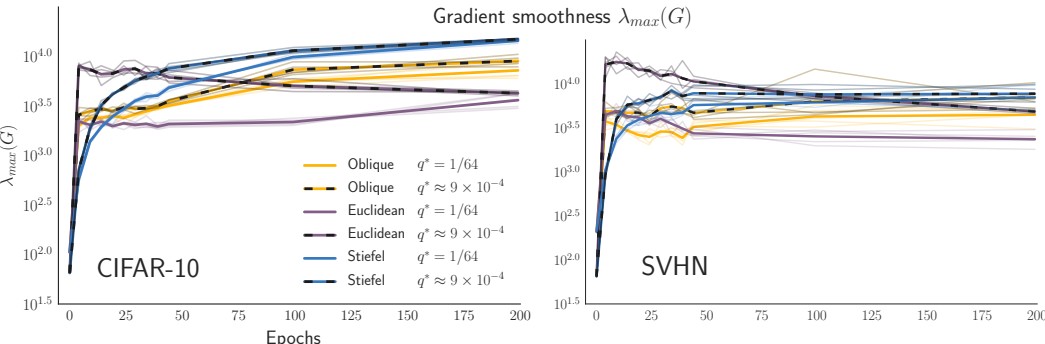

Figure 3: For manifold constrained networks, gradient smoothness is not predictive of optimization rate. Euclidean networks with a low initial $\lambda_{\max}(\mathbf{G})$ rapidly become less smooth, whereas Euclidean networks with a larger $\lambda_{max}(\mathbf{G})$ remain relatively smoother. Notably, the Euclidean network with $q^* = 1/64$ has almost an order of magnitude smaller $\lambda_{\max}(\mathbf{G})$ than the Stiefel and Oblique networks, but reduces training loss at a slower rate.

## 4 DISCUSSION

Critical orthogonal initializations have proven tremendously successful in rapidly training very deep neural networks (Pennington et al., 2017; Chen et al., 2018; Pennington et al., 2018; Xiao et al., 2018). Despite their elegant derivation drawing on methods from free probability and mean field theory, they did not offer a clear optimization perspective on the mechanisms driving their success. With this work we complement the understanding of critical orthogonal initializations by showing that the maximum eigenvalue of the Fisher information matrix, and consequentially the local gradient smoothness is proportional to the maximum singular value of the input-output Jacobian. This gives an information geometric account of why the step size and training speed depend on $q^*$ via its effect on $\mathbb{E}\left[s_{\max}(\mathbf{J}_{\mathbf{h}^1}^{\mathbf{h}^L})\right]$. We observed in numerical experiments that the paradoxical results reported in (Pennington et al., 2017) whereby training speed and generalization attains a maximum for $q^* = 10^{-0.5}$ can potentially be explained by a rapid increase of the maximum eigenvalue of the FIM during training for the networks initialized with Jacobians closer to being isometric (i.e., smaller $q^*$). This increase effectively limits the learning rate during the early phase of optimization

and highlights the need to analyze the trajectories of training rather than just initializations. Notably these effects persist under extensive search for the optimal learning rate, indicating the increased difficulty in training highly isometric neural networks reflects a qualitative change of the optimization problem. We conjecture that this phenomenon relates to the stability of descent direction in response to small parameter perturbations. An analysis of the cubic term (Riemmanian connection) would give insight into the initial change in curvature at initialization, however that currently seems infeasible both numerically and analytically. In lieu of that, let us consider the stability of neural networks linearized about their initial parameters. A number of works on overparametrized networks have derived stability conditions for the linearization both in infinitesmal regimes and also over potentially infinitely many gradient updates (Chizat et al., 2019; Lee et al., 2019). The crucial property governing this stability is ensuring that the relative change in parameters is small in proportion to the norm of the Gram matrix of the gradients of the networks. The latter curiously has a spectrum that coincides with that of the Fisher information matrix under mean-squared loss (see A.7). Other arguments about the stability of the linearization have proposed bounds that depend non-trivially on a function of both the largest and smallest eigenvalues of this Gram matrix (Lee et al., 2019). It is therefore tempting to understand the behavior of the lowest eigenvalue of this curvature matrix. Lower bounds on the smallest eigenvalue are typically much harder to obtain, however Karakida et al. (2019) showed that the smallest eigenvalue of the Fisher scales roughly reciprocally with the maximal one. This might imply that a low condition number $\frac{\lambda_{\max}(\mathbf{G}_0)}{\lambda_{\max}(\mathbf{G}_0)}$ may be undesirable, and a degree of anisotropy is necessary for the Fisher information matrix at initialization to be predictive of training performance. We observe that orthogonal *linear* networks experience a similar rapid increase in the maximum eigenvalue of their FIM, while having an FIM condition number that is $\mathcal{O}(1)$, up to applying similarity transform to the input data (see Appendix A.8).

Finally, we compared manifold constrained networks with the Euclidean network, each evaluated with two initial values of $q^*$. From these experiments we draw the conclusion that manifold constrained networks are less sensitive to the initial strong smoothness, unlike their Euclidean counterparts. Furthermore, we observe that the rate at which Stiefel and Oblique networks decrease training loss is not dependent on their gradient smoothness, a result which is consistent with the recent analysis of (Kohler et al., 2019).

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

# A APPENDIX

## A.1 BLOCK GERSHGORIN THEOREM

In Section 3, we considered a block diagonal approximation to the Fisher information matrix and derived an upper bound on the spectral norm for all the blocks. Using the properties of the off-diagonal blocks, we can get a more accurate estimate of the maximal eigenvalue of the Fisher information might be. First, let us consider an arbitrarily partitioned matrix $\mathbf{A} \in \mathbb{R}^{N \times N}$, with spectrum $\lambda(\mathbf{A})$ The partitioning is done with respect to the set

$$\pi = \{p_j\}_{j=0}^{L} \tag{18}$$

with the elements of the set satisfying $0 < p_1 < p_2 < \ldots < p_L = N$. Then each block of the matrix $A_{i,j}$ is a potentially rectangular matrix in $\mathbb{R}^{(p_i - p_{i-1}) \times (p_j - p_{j-1})}$. We assume that $\mathbf{A}_{i,i}$ is self-adjoint for all $i$.

Let us define a disk as

$$C(c,r) \triangleq \{\lambda : \|c - \lambda\| \leq r\}. \tag{19}$$

The theorem as presented in (Tretter, 2008) shows that the eigenvalues of $\lambda(\mathbf{A})$ are contained in a union of Gershgorin disks defined as follows

$$\lambda(\mathbf{A}) \subset \bigcup_{i=1}^{L} \left\{ \bigcup_{k=1}^{p_i - p_{i-1}} C\left( \lambda_k(\mathbf{A}_{ii}), \sum_{j=1, j \neq i}^{L} s_{\max}(\mathbf{A}_{i,j}) \right) \right\} \tag{20}$$

where the inner union is over a set disks for each eigenvalue of the block diagonal $\mathbf{A}_{i,i}$ while the outer union is over the $L$ blocks in $\mathbf{A}$. The radius of the disk is constant for every eigenvalue in the $i^{\text{th}}$ diagonal block $\mathbf{A}_{i,i}$ and is given by the sum of singular values of the off diagonal blocks. Therefore, the largest eigenvalue of $\mathbf{A}$ lies in

$$\lambda_{\max}(\mathbf{A}) \subset \bigcup_{i=1}^{L} C\left( \lambda_{\max}(\mathbf{A}_{ii}), \sum_{j=1, j \neq i}^{L} s_{\max}(\mathbf{A}_{i,j}) \right) \tag{21}$$

Now suppose $\mathbf{A}$ is positive semidefinite. This suggests a strategy for upper-bounding the maximal eigenvalue of $\mathbf{A}$— picking the disk containing the largest (non-negative) value will upper-bound the maximal eigenvalue of $\mathbf{A}$:

$$\lambda_{\max}(\mathbf{A}) \leq \max_i \left( \lambda_{\max}(\mathbf{A}_{ii}) + \sum_{j=1, j \neq i}^{L} s_{\max}(\mathbf{A}_{i,j}) \right) \tag{22}$$

## A.2 DERIVATION OF THE EXPECTED SINGULAR VALUES

We present a derivation of the bounds presented in Lemma 3. The singular value bound for the weight-to-weight blocks is

$$\sigma_{\max}\left(\mathbf{G}_{\text{vec}(\mathbf{W}^1), \text{vec}(\mathbf{W}^\beta)}\right) = \mathbb{E}\left[\sigma_{\max}\left(\phi(h)\mathbb{1}\,\mathbf{x}^{0\top}\right)\right] \otimes \mathbb{E}\left[\sigma_{\max}\left(\mathbf{J}_{\mathbf{h}^1}^{\mathbf{h}^g\top}\mathbf{H}_g\mathbf{J}_{\mathbf{h}^\beta}^{\mathbf{h}^g}\right)\right] \tag{23}$$

$$= \sqrt{N^\beta}\,|\mathbb{E}[\phi(h)]|\,\left\|\mathbb{E}\left[\mathbf{x}^0\right]\right\|_2 \mathbb{E}\left[\sigma_{\max}\left(\mathbf{J}_{\mathbf{h}^1}^{\mathbf{h}^g\top}\mathbf{H}_g\mathbf{J}_{\mathbf{h}^\beta}^{\mathbf{h}^g}\right)\right] \tag{24}$$

where $\phi(h)\mathbb{1}$ is a vector of ones times $\phi(h)$

$$\leq \sqrt{N^\beta}\,|\mathbb{E}[\phi(h)]|\,\left\|\mathbb{E}\left[\mathbf{x}^0\right]\right\|_2 \mathbb{E}\left[\sigma_{\max}\left(\mathbf{J}_{\mathbf{h}^1}^{\mathbf{h}^g\top}\right)\right] \mathbb{E}\left[\sigma_{\max}(\mathbf{H}_g)\right] \mathbb{E}\left[\sigma_{\max}\left(\mathbf{J}_{\mathbf{h}^\beta}^{\mathbf{h}^g}\right)\right] \tag{25}$$

The singular value bound for the weight-to-bias blocks is

$$\sigma_{\max}\left(\mathbf{G}_{\text{vec}(\mathbf{W}^1), b^\beta}\right) \leq \mathbb{E}\left[\sigma_{\max}\left(\mathbf{x}^{0\top} \otimes \mathbf{I}\right)\right] \mathbb{E}\left[\sigma_{\max}\left(\mathbf{J}_{\mathbf{h}^\alpha}^{\mathbf{h}^g\top}\mathbf{H}_g\mathbf{J}_{\mathbf{h}^\beta}^{\mathbf{h}^g}\right)\right] \tag{26}$$

$$= \left\|\mathbb{E}\left[\mathbf{x}^0\right]\right\|_2 \left(\mathbb{E}\left[\sigma_{\max}\mathbf{J}_{\mathbf{h}^1}^{\mathbf{h}^g\top}\mathbf{H}_g\mathbf{J}_{\mathbf{h}^\beta}^{\mathbf{h}^g}\right)\right] \tag{27}$$

$$\leq \left\|\mathbb{E}\left[\mathbf{x}^0\right]\right\|_2 \mathbb{E}\left[\sigma_{\max}\left(\mathbf{J}_{\mathbf{h}^1}^{\mathbf{h}^g\top}\right)\right] \mathbb{E}\left[\sigma_{\max}(\mathbf{H}_g)\right] \mathbb{E}\left[\sigma_{\max}\left(\mathbf{J}_{\mathbf{h}^\beta}^{\mathbf{h}^g}\right)\right] \tag{28}$$

$$\sigma_{\max}\left(\mathbf{G}_{b^1,\mathrm{vec}(\mathbf{W}^\beta)}\right) \leq \mathbb{E}\left[\sigma_{\max}\left(\mathbf{x}^{\beta-1\top} \otimes \mathbf{I}\right)\right] \tag{29}$$

$$= \sqrt{N^\beta}\,|\mathbb{E}\left[\phi(h)\right]|\,\mathbb{E}\left[\sigma_{\max}\left(\mathbf{J}_{\mathbf{h}^1}^{\mathbf{h}^g\top}\mathbf{H}_g\mathbf{J}_{\mathbf{h}^\beta}^{\mathbf{h}^g}\right)\right] \tag{30}$$

$$\leq \sqrt{N^\beta}\,|\mathbb{E}\left[\phi(h)\right]|\,\mathbb{E}\left[\sigma_{\max}\left(\mathbf{J}_{\mathbf{h}^1}^{\mathbf{h}^g\top}\right)\right]\mathbb{E}\left[\sigma_{\max}\left(\mathbf{H}_g\right)\right]\mathbb{E}\left[\sigma_{\max}\left(\mathbf{J}_{\mathbf{h}^\beta}^{\mathbf{h}^g}\right)\right] \tag{31}$$

The singular value bound for the weight-to-bias blocks is

$$\sigma_{\max}\left(\mathbf{G}_{b^1,b^\beta}\right) = \mathbb{E}\left[\sigma_{\max}\left(\mathbf{J}_{\mathbf{h}^1}^{\mathbf{h}^g\top}\mathbf{H}_g\mathbf{J}_{\mathbf{h}^\beta}^{\mathbf{h}^g}\right)\right] \tag{32}$$

$$\leq \mathbb{E}\left[\sigma_{\max}\left(\mathbf{J}_{\mathbf{h}^1}^{\mathbf{h}^g\top}\right)\right]\mathbb{E}\left[\sigma_{\max}\left(\mathbf{H}_g\right)\right]\mathbb{E}\left[\sigma_{\max}\left(\mathbf{J}_{\mathbf{h}^\beta}^{\mathbf{h}^g}\right)\right] \tag{33}$$

### A.3 MONTECARLO ESTIMATE OF MAXIMUM EIGENVALUE OF THE HESSIAN OF THE OUTPUT LAYER FOR 10 WAY SOFTMAX CLASSIFICATION

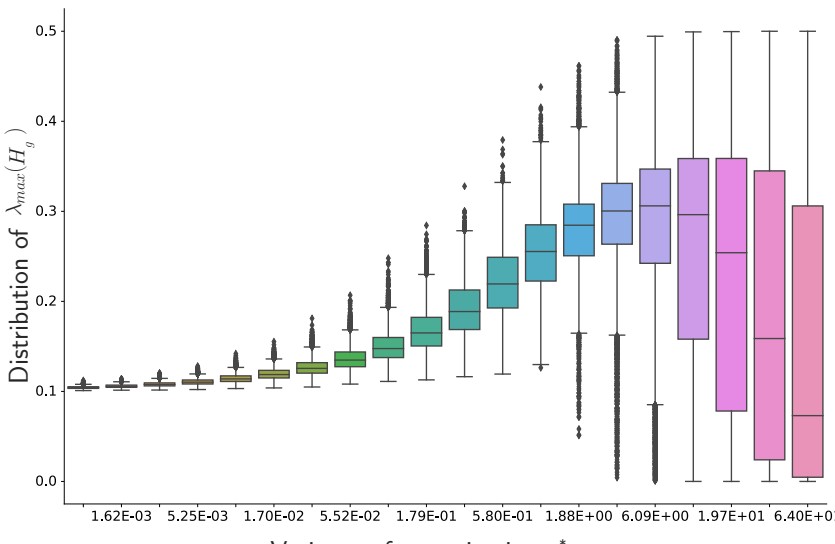

Figure 4: **Distribution of $\lambda_{\max}(\mathbf{H}_g)$ as a function of** $q^*$: In general, increasing the variance of the distribution of $h^g$ does not result in a monotonic increase in the spectral radius of the Hessian of the GLM layer. We plot the distribution of the maximum eigenvalues as a function of the variance of the softmax layer obtained from factorizing 10,000 random matrices.

### A.4 PROOF OF THEOREM

Using the results from Section A.2, we observe that all the terms off diagonal for the diagonal block $\mathbf{G}_{\mathrm{vec}(\mathbf{W}^1),\mathrm{vec}(\mathbf{W}^1)}$ are of the form

$$|\mathbb{E}\left[\phi(h)\right]|\,\left\|\mathbb{E}\left[\mathbf{x}^0\right]\right\|_2\mathbb{E}\left[\sigma_{\max}\left(\mathbf{J}_{\mathbf{h}^1}^{\mathbf{h}^g\top}\right)\right]\mathbb{E}\left[\sigma_{\max}\left(\mathbf{H}_g\right)\right]\mathbb{E}\left[\sigma_{\max}\left(\mathbf{J}_{\mathbf{h}^\beta}^{\mathbf{h}^g}\right)\right] \tag{34}$$

and

$$\left\|\mathbb{E}\left[\mathbf{x}^0\right]\right\|_2\mathbb{E}\left[\sigma_{\max}\left(\mathbf{J}_{\mathbf{h}^1}^{\mathbf{h}^g\top}\right)\right]\mathbb{E}\left[\sigma_{\max}\left(\mathbf{H}_g\right)\right]\mathbb{E}\left[\sigma_{\max}\left(\mathbf{J}_{\mathbf{h}^\beta}^{\mathbf{h}^g}\right)\right] \tag{35}$$

while the diagonal term is

$$\lambda_{\max}(\mathbb{E}\left[\mathbf{x}^{0\top}\mathbf{x}^0\right])\mathbb{E}\left[\sigma_{\max}\left(\mathbf{J}_{\mathbf{h}^1}^{\mathbf{h}^g\top}\right)\right]\mathbb{E}\left[\sigma_{\max}\left(\mathbf{H}_g\right)\right]\mathbb{E}\left[\sigma_{\max}\left(\mathbf{J}_{\mathbf{h}^\beta}^{\mathbf{h}^g}\right)\right] \tag{36}$$

then by Gershgorin theorem the eigenvalues for the $\mathbf{G}_{\mathrm{vec}(\mathbf{W}^1),\mathrm{vec}(\mathbf{W}^1)}$ are bounded by

$$\lambda_{\max}(\mathbf{G}_{\mathrm{vec}(\mathbf{W}^1),\mathrm{vec}(\mathbf{W}^1)}) \leq \mathbb{E}\left[\sigma_{\max}\left(\mathbf{J}_{\mathbf{h}^1}^{\mathbf{h}^g\top}\right)\right]\mathbb{E}\left[\sigma_{\max}\left(\mathbf{H}_g\right)\right]$$

$$\times\left[\lambda_{\max}(\mathbb{E}\left[\mathbf{x}^{0\top}\mathbf{x}^0\right])\mathbb{E}\left[\sigma_{\max}\left(\mathbf{J}_{\mathbf{h}^1}^{\mathbf{h}^g\top}\right)\right] + \sum_{\beta}\left(1 + |\mathbb{E}[\phi(h)]|\right)\left\|\mathbb{E}\left[\mathbf{x}^0\right]\right\|_2\mathbb{E}\left[\sigma_{\max}\left(\mathbf{J}_{\mathbf{h}^\beta}^{\mathbf{h}^g}\right)\right]\right] \tag{37}$$

Similarly, for the $\mathbf{G}_{b^1,b^1}$ block, we observe that

$$
\lambda_{\max}(\mathbf{G}_{b^1,b^1}) \leq \left( \mathbb{E}\left[ \sigma_{\max}\left( \mathbf{J}_{\mathbf{h}^1}^{\mathbf{h}^g \top} \right) \right] \mathbb{E}\left[ \sigma_{\max}\left( \mathbf{H}_g \right) \right] \right) \times
$$
$$
\times \left[ \sum_\beta \left( 1 + \sqrt{N^\beta} |\mathbb{E}[\phi(h)]| \right) \left\| \mathbb{E}\left[ \mathbf{x}^0 \right] \right\|_2 \mathbb{E}\left[ \sigma_{\max}\left( \mathbf{J}_{\mathbf{h}^\beta}^{\mathbf{h}^g} \right) \right] \right]
$$

(38)

## A.5 ADDITIONAL FIGURES

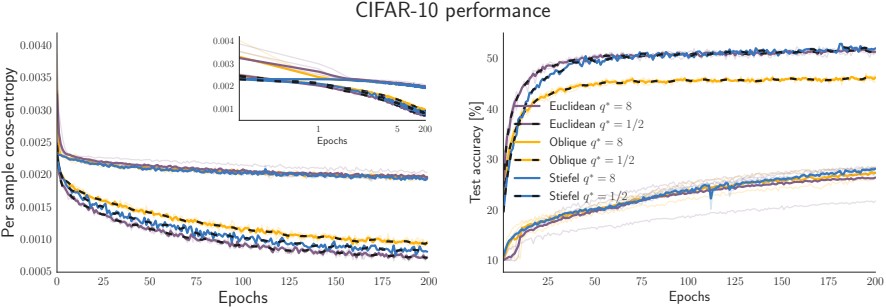

Figure 5: Training performance for networks with $q^* = \frac{1}{2}$ and $q^* = 8$. The behavior of the training loss as well as the validation accuracy is consistent with the observations that for a large range of parameters $q^*$ the manifold constrained networks are insensitive to initialization and gradient smoothness.

## A.6 MANIFOLD OPTIMIZATION

The potentially non-convex constraint set constitutes a Riemannian manifold, when it is locally isomorphic to $\mathbb{R}^n$, differentiable and endowed with a suitable (Riemannian) metric, which allows us to measure distances in the tangent space and consequentially also define distances on the manifold. There is considerable freedom in choosing a Riemannian metric; here we consider the metric inherited from the Euclidean embedding space which is defined as $\langle \mathbf{W}, \mathbf{W}' \rangle \triangleq \text{Tr}(\mathbf{W}'^\top \mathbf{W})$. To optimize a cost function with respect to parameters lying in a non-Euclidean manifold we must define a descent direction. This is done by defining a manifold equivalent of the directional derivative. An intuitive approach replaces the movement along a vector $\mathbf{t}$ with movement along a geodesic curve $\gamma(t)$, which lies in the manifold and connects two points $\mathbf{W}, \mathbf{W}' \in \mathcal{M}$ such that $\gamma(0) = \mathbf{W}$, $\gamma(1) = \mathbf{W}'$. The derivative of an arbitrary smooth function $f(\gamma(t))$ with respect to $t$ then defines a tangent vector for each $t$.

**Tangent vector** $\xi_{\mathbf{W}}$ is a tangent vector at $\mathbf{W}$ if $\xi_{\mathbf{W}}$ satisfies $\gamma(0) = \mathbf{W}$ and

$$
\xi_{\mathbf{W}} \triangleq \left. \frac{\mathrm{d}f(\gamma(t))}{\mathrm{d}t} \right|_{t=0} \triangleq \gamma'(0)f
$$

(39)

The set of all tangents to $\mathcal{M}$ at $\mathbf{W}$ is referred to as the tangent space to $\mathcal{M}$ at $\mathbf{W}$ and is denoted by $T_{\mathbf{W}}\mathcal{M}$. The geodesic importantly is then specified by a constant velocity curve $\gamma''(t) = 0$ with initial velocity $\xi_{\mathbf{W}}$. To perform a gradient step, we must then move along $\xi_{\mathbf{W}}$ while respecting the manifold constraint. This is achieved by applying the exponential map defined as $\text{Exp}_{\mathbf{W}}(\xi_{\mathbf{W}}) \triangleq \gamma(1)$, which moves $\mathbf{W}$ to another point $\mathbf{W}'$ along the geodesic. While certain manifolds, such as the Oblique manifold, have efficient closed-form exponential maps, for general Riemannian manifolds, the computation of the exponential map involves numerical solution to a non-linear ordinary differential equation (Absil et al., 2007). An efficient alternative to numerical integration is given by an orthogonal projection onto the manifold. This projection is formally referred to as a retraction $\text{Rt}_{\mathbf{W}} : T_{\mathbf{W}}\mathcal{M} \to \mathcal{M}$.

Finally, gradient methods using Polyak (heavy ball) momentum (e.g. ADAM (Kingma & Ba, 2015)) require the iterative updating of terms which naturally lie in the tangent space. The parallel translation $\mathcal{T}_\zeta(\xi) : T\mathcal{M} \bigoplus T\mathcal{M} \to T\mathcal{M}$ generalizes vector composition from Euclidean to non-Euclidean manifolds, by moving the tangent $\xi$ along the geodesic with initial velocity $\zeta \in \mathcal{T}$ and endpoint $\mathbf{W}'$, and then projecting the resulting vector onto the tangent space $T_{\mathbf{W}'}\mathcal{M}$. As with the exponential map,

parallel transport $\mathcal{T}$ may require the solution of non-linear ordinary differential equation. To alleviate the computational burden, we consider *vector transport* as an effective, projection-like solution to the parallel translation problem. We overload the notation and also denote it as $\mathcal{T}$, highlighting the similar role that the two mappings share. Technically, the geodesics and consequentially the exponential map, retraction as well as transport $\mathcal{T}$ depend on the choice of the Riemannian metric. Putting the equations together the updating scheme for Riemannian stochastic gradient descent on the manifold is

$$\mathbf{W}_{t+1} = \Pi_{\mathbf{W}_t}(-\eta_t \operatorname{grad} f) \tag{40}$$

where $\Pi$ is either the exponential map $\operatorname{Exp}$ or the retraction $\operatorname{Rt}$ and $\operatorname{grad} f$ is the gradient of the function $f(\mathbf{W})$ lying in the tangent space $T_{\mathbf{W}}\mathcal{M}$.

### A.6.1 OPTIMIZING OVER THE OBLIQUE MANIFOLD

(Cho & Lee, 2017) proposed an updating scheme for optimizing neural networks where the weights of each layer are constrained to lie in the oblique manifold $\operatorname{Ob}(p, n)$. Using the fact that the manifold itself is a product of $p$ unit-norm spherical manifolds, they derived an efficient, closed-form Riemannian gradient descent updating scheme. In particular the optimization simplifies to the optimization over $\operatorname{Ob}(1, n)$ for each column $\mathbf{w}_{i \in \{1, \dots, p\}}$ of $\mathbf{W}$.

**Oblique gradient**   The gradient $\operatorname{grad} f$ of the cost function f with respect to the weights lying in $\operatorname{Ob}(1, n)$ is given as a projection of the Euclidean gradient $\operatorname{Grad} f$ onto the tangent at $\mathbf{w}$

$$\operatorname{grad} f = \operatorname{Grad} f - (\mathbf{w}^\top \operatorname{Grad} f)\mathbf{w} \tag{41}$$

**Oblique exponential map**   The exponential map $\operatorname{Exp}_{\mathbf{w}}$ moving $\mathbf{w}$ to $\mathbf{w}'$ along a geodesic with initial velocity $\xi_{\mathbf{w}}$

$$\operatorname{Exp}_{\mathbf{w}} = \xi_{\mathbf{w}} \cos(\|\mathbf{w}\|) + \frac{\mathbf{w}}{\|\mathbf{w}\|} \sin(\|\mathbf{w}\|) \tag{42}$$

**Oblique parallel translation**   The parallel translation $\mathcal{T}$ moves the tangent vector $\xi_{\mathbf{w}}$ along the geodesic with initial velocity $\zeta_{\mathbf{w}}$

$$\mathcal{T}_{\zeta_{\mathbf{w}}}(\xi_{\mathbf{w}}) = \xi_{\mathbf{w}} - \frac{\zeta_{\mathbf{w}}}{\|\zeta_{\mathbf{w}}\|}\left((1 - \cos(\|\zeta_{\mathbf{w}}\|)) + \mathbf{w}\sin(\|\zeta_{\mathbf{w}}\|)\right)\frac{\zeta_{\mathbf{w}}}{\|\zeta_{\mathbf{w}}\|}^\top \xi_{\mathbf{w}}$$

**Orthogonal penalty**   which enforces the weight matrices to be near orthogonal.

$$\rho(\lambda, \mathbf{W}) = \frac{\lambda}{2}\left\|\mathbf{W}^\top \mathbf{W} - \mathbf{I}\right\|_F^2 \tag{43}$$

### A.6.2 OPTIMIZING OVER THE STIEFEL MANIFOLD

Optimization over Stiefel manifolds in the context of neural networks has been studied by (Harandi & Fernando, 2016; Wisdom et al., 2016; Vorontsov et al., 2017). Unlike (Wisdom et al., 2016; Vorontsov et al., 2017) we propose the parametrization using the Euclidean metric, which results in a different definition of vector transport.

**Stiefel gradient**   The gradient $\operatorname{grad} f$ of the cost function f with respect to the weights lying in $\operatorname{St}(p, n)$ is given as a projection of the Euclidean gradient $\operatorname{Grad} f$ onto the tangent at $\mathbf{W}$ (Edelman et al., 1999; Absil et al., 2007)

$$\operatorname{grad} f = (\mathbf{I} - \mathbf{W}\mathbf{W}^\top)\operatorname{Grad} f + \frac{1}{2}\mathbf{W}\left(\mathbf{W}^\top \operatorname{Grad} f - \operatorname{Grad} f^\top \mathbf{W}\right)$$

**Stiefel retraction**   The retraction $\operatorname{Rt}_{\mathbf{W}}(\xi_{\mathbf{W}})$ for the Stiefel manifold is given by the Q factor of the QR decomposition (Absil et al., 2007).

$$\operatorname{Rt}_{\mathbf{W}}(\xi_{\mathbf{W}}) = \operatorname{qf}(\mathbf{W} + \xi_{\mathbf{W}}) \tag{44}$$

**Stiefel vector transport** The vector transport $\mathcal{T}$ moves the tangent vector $\xi_{\mathbf{w}}$ along the geodesic with initial velocity $\zeta_{\mathbf{w}}$ for $\mathbf{W} \in \mathrm{St}(p, n)$ endowed with the Euclidean metric.

$$\mathcal{T}_{\zeta_{\mathbf{w}}}(\xi_{\mathbf{w}}) = \left(\mathbf{I} - \mathbf{Y}\mathbf{Y}^{\top}\right)\xi_{\mathbf{W}} + \frac{1}{2}\mathbf{Y}\left(\mathbf{Y}^{\top}\xi_{\mathbf{W}} - \xi_{\mathbf{W}}^{\top}\mathbf{Y}\right) \tag{45}$$

where $\mathbf{Y} \triangleq \mathrm{Rt}_{\mathbf{W}}(\zeta_{\mathbf{W}})$. It is easy to see that the transport $\mathcal{T}$ consists of a retraction of tangent $\zeta_{\mathbf{W}}$ followed by the orthogonal projection of $\eta_{\mathbf{W}}$ at $\mathrm{Rt}_{\mathbf{W}}(\zeta_{\mathbf{W}})$.

### A.6.3 OPTIMIZING OVER NON-COMPACT MANIFOLDS

The critical weight initialization yielding a singular spectrum of the Jacobian tightly concentrating on $1$ implies that a substantial fraction of the pre-activations lie in expectation in the linear regime of the squashing nonlinearity and as a consequence the network acts quasi-linearly. To relax this constraint during training we allow the scales of the manifold constrained weights to vary. We chose to represent the weights as a product of a scaling diagonal matrix and a matrix belonging to the manifold. Then the optimization of each layer consists in the optimization of the two variables in the product. In this work we only consider isotropic scalings, but the method generalizes easily to the use of any invertible square matrix.

It is interesting to contrast our approach with projected gradient descent with a spectral prox function, or penalizing the Jacobian norm through back propagation. PGD with projections onto compact spectral balls requires the user to pre-specify the desired spectral radius. On the other hand, both our approach as well as penalizing the Jacobian norm can be thought of penalizing decreases in function smoothness in an adaptive way. Finally, our approach is naturally amenable to incorporating a data-dependent penalty — this would allow us to smoothly vary the spectral radia of the weight matrices.

### A.7 FIM AND NTK HAVE THE SAME SPECTRUM

The empirical Neural Tangent Kernel (NTK) is defined as

$$\hat{\Theta}_{t,i,j} \triangleq \mathbf{J}_{\theta}^{\mathbf{h}^{g}}\mathbf{J}_{\theta}^{\mathbf{h}^{g}\top} \tag{46}$$

which gives a $N^{g}|\mathcal{D}|$ by $N^{g}|\mathcal{D}|$ kernel matrix. By comparison the empirical Fisher information matrix with a Gaussian likelihood is

$$\sum_{i \in |\mathcal{D}|} \mathbf{J}_{\theta}^{h^{g}\top}\mathbf{J}_{\theta}^{h^{g}} \tag{47}$$

To see that the spectra of these two coincide consider the third order tensor underlying both $\mathbf{J}_{\mathbf{h}^{1}i}^{\mathbf{h}^{g}}$ for $i \in 1 \ldots |\mathcal{D}|$, additionally consider and unfolding $\mathbf{A}$ with dimensions $|\theta|$ by $N^{g}|\mathcal{D}|$; i.e. we construct a matrix with dimension of number of parameters by number of outputs times number of data points. Then

$$\mathbf{G} = \mathbf{A}^{\top}\mathbf{A} \tag{48}$$

$$\hat{\Theta} = \mathbf{A}\mathbf{A}^{\top} \tag{49}$$

$$\tag{50}$$

and their spectra trivially coincide.

## A.8 LINEAR NEURAL NETWORKS WITH ORTHOGONAL INITIALIZATIONS

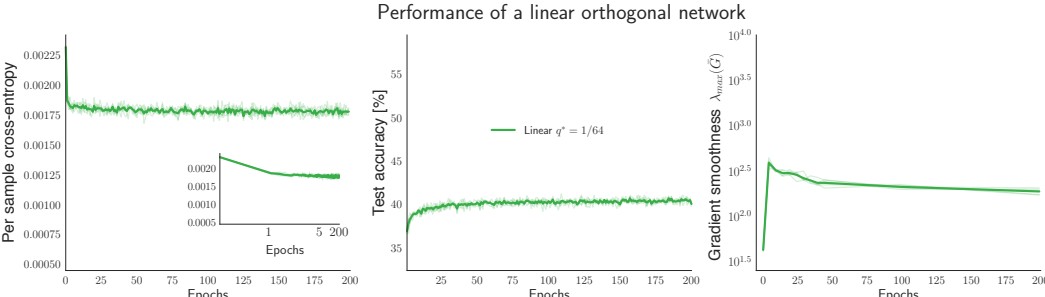

Figure 6: Linear neural networks with orthogonal initializations increase rapidly in their maximum eigenvalue of the empirical Fisher information, similar to their non-linear, nearly isometric counterparts.

