# OpenReview forum: "Information Geometry of Orthogonal Initializations and Training"
_ICLR.cc/2020/Conference — Accept (Poster)_

### Official Review · AnonReviewer3 · 2019-10-22
**Official Blind Review #3**

**Rating:** 6

**Review:**

This paper analyzes the connection between the spectrum of the layer-to-layer or input-output Jacobian matrices and the spectrum of the Fisher information matrix / Neural Tangent Kernel. By bounding the maximum eigenvalue of the Fisher in terms of the maximum squared singular value of the input-output Jacobian, this paper provides a partial explanation for the successful initialization procedures obeying "dynamical isometry". By additionally investigating optimization on the orthogonal weight manifold, this paper sheds light on the important of maintaining spectral uniformity throughout training.  These two analyses help fill in important gaps in the understanding of initialization, dynamical isometry, and the training of deep neural networks. For these reasons I recommend this paper for acceptance.

There are two aspects of the paper that could nevertheless use some clarification and improvement. First, unless I missed something, this paper does not provide any bounds on the condition number or the minimum eigenvalue of the Fisher or the NTK. It seems like the main arguments only depend on the maximum eigenvalues. Generally, I think the insights into the maximum eigenvalues are useful and important on their own, but perhaps some additional discussion up front clarifying which results were derived theoretically and which were observed empirically could be useful.

Second, it should be noted that the the networks trained in the experiments are likely in a regime that is well outside the NTK regime, in two important ways: the dataset is large compared to the width and the optimal learning rates may be large as well.

Overall, I think this is a good paper that adds important insights into the study of initialization, local geometry, and their effects on training speed.

**Experience Assessment:**

I have published in this field for several years.

**Review Assessment: Checking Correctness Of Derivations And Theory:**

I assessed the sensibility of the derivations and theory.

**Review Assessment: Checking Correctness Of Experiments:**

I assessed the sensibility of the experiments.

**Review Assessment: Thoroughness In Paper Reading:**

I read the paper at least twice and used my best judgement in assessing the paper.

---

> ### Author Response · Authors · 2019-11-15
> **RE: Reviewer 3**
>
> We thank the reviewer for their thoughtful comments.
>
> We agree that understanding the smallest eigenvalue of the Fisher information matrix (FIM) would be invaluable. We are however unfamiliar with any ways of directly lower-bounding that quantity. Under somewhat specific conditions Karakida et al [1] that as $\lambda_{\max}(G) \to \infty$ then $\lambda_{\min} \to 0$. This result hinges on the assumptions that weight matrices are distributed as $ W_{i,j} \sim \mathcal{N} (0, \sigma^2_{l}/ M_{l-1})$ where M_{l-1} is the number of neurons in layer $l-1$ then the mean eigevalue is  $|G|_{Fro}$ and the mean eigenvalue is $ \frac{1}{P} |G|_{Fro}$ where is the total number of parameters. Importantly for Gaussian weights the maximum eigenvalue grows as $\mathcal{O}(M)$ but the mean eigenvalue is proportional to $\mathcal{O}(1/M)$ and a fortiori the minimal eigenvalue must decrease at least as fast.
> We agree that an upfront discussion of this special case would give the reader a better understanding of the context and potential implications. In a similar vein, we believe that adding a disclaimer that estimating the condition number is generally numerically unstable.
>
> We whole-heartedly agree that the some of the conditions necessary for the NTK may not hold in our experiments, and therefore ammended the manuscript appropriately. The authors in [2,3] consider widths of up to 10,000, compared to ours 400 neuron networks. However, [4] consider networks of similar width to ours.
> Moreover, it has been pointed out that gradient flow, like any differential equation admits a convergent forward Euler discretization, provided the norm of the eigenvalues of the one-step update obey $ |1 - lambda| \le 1 $. We did not check that these conditions are met; we automatically chose learning rates that produced the highest validation accuracy after 50 epochs. We show the learning rates at the bottom of this response, and will release the entire entire set of hyperparameters as JSON files in the github repo. Moreover, in the amended version of the document we show will that our conjectured explanation hold with the above mentioned qualifications.
>
> Example learning rates for CIFAR-10 and h0=0.0009236716627770724 using ADAM
>
>     "config" : {
>         "h0" : 0.0009236716627770724,
>         "manifold" : "stiefel",
>         "learning_rate_euclidean" : 0.00009405111292996846,
>         "learning_rate_manifold" : 0.000016915446998604953,
>         "learning_rate_scale" : 0.00005043477108079471,
>         "omega" : NumberInt(0),
>         "weight_decay" : NumberInt(0)
>     }
>
>     "config" : {
>         "h0" : 0.0009236716627770724,
>         "manifold" : "euclidean",
>         "learning_rate_euclidean" : 0.00001434368393061965,
>         "learning_rate_manifold" : 0.0,
>         "learning_rate_scale" : 0.0,
>         "omega" : NumberInt(0),
>         "weight_decay" : NumberInt(0)
>     }
>
>     "config" : {
>         "h0" : 0.0009236716627770724,
>         "manifold" : "oblique",
>         "learning_rate_euclidean" : 0.000021724540834816572,
>         "learning_rate_manifold" : 0.00000997854628342336,
>         "learning_rate_scale" : 0.00007832919428972671,
>         "omega" : 0.0007680321077268013,
>         "weight_decay" : NumberInt(0)
>     }
> learning rates for CIFAR-10 and h0=1/64 using ADAM
>     "config" : {
>         "h0" : 0.015625,
>         "learning_rate_euclidean" : 0.000008289977483912705,
>         "learning_rate_manifold" : 0.000026997346360225043,
>         "learning_rate_scale" : 0.00007146727754603447,
>         "manifold" : "stiefel",
>         "omega" : NumberInt(0)
>     }
>
>     "config" : {
>         "h0" : 0.015625,
>         "learning_rate_euclidean" : 0.000023036386339653812,
>         "learning_rate_manifold" : 0.0,
>         "learning_rate_scale" : 0.0,
>         "manifold" : "euclidean",
>         "omega" : NumberInt(0)
>     }
>
>     "config" : {
>         "h0" : 0.015625,
>         "learning_rate_euclidean" : 0.00006560909241433484,
>         "learning_rate_manifold" : 0.000020133287475822923,
>         "learning_rate_scale" : 0.00008723072785306676,
>         "manifold" : "oblique",
>         "omega" : 0.00026312093948723203,
>         "weight_decay" : NumberInt(0)
>     }

---

> > ### Author Response · Authors · 2019-11-15
> > **references**
> >
> >
> > [1]R. Karakida, S. Akaho, and S. Amari, “Universal Statistics of Fisher Information in Deep Neural Networks: Mean Field Approach,” in The 22nd International Conference on Artificial Intelligence and Statistics, 2019, pp. 1032–1041.
> > [2]A. Jacot, F. Gabriel, and C. Hongler, “Neural Tangent Kernel: Convergence and Generalization in Neural Networks,” in Advances in Neural Information Processing Systems 31, S. Bengio, H. Wallach, H. Larochelle, K. Grauman, N. Cesa-Bianchi, and R. Garnett, Eds. Curran Associates, Inc., 2018, pp. 8571–8580.
> > [3]J. Lee, L. Xiao, S. S. Schoenholz, Y. Bahri, J. Sohl-Dickstein, and J. Pennington, “Wide Neural Networks of Any Depth Evolve as Linear Models Under Gradient Descent,” p. 18.
> > [4]S. Hayou, A. Doucet, and J. Rousseau, “On the Impact of the Activation function on Deep Neural Networks Training,” in International Conference on Machine Learning, 2019, pp. 2672–2680.

---

### Official Review · AnonReviewer1 · 2019-10-22
**Official Blind Review #1**

**Rating:** 8

**Review:**

This paper formulates a connection between the Fisher information matrix (FIM) and the spectral radius of the input-output Jacobian in neural networks. This results derive the eigenvalues' bound to theoretically study the convergence of several networks.  Here the upper bound further improves the upper bound of FIM derived in (Karakida et al., 2018).
This is a very interesting and useful direction of applying information matrices to study the initialization of deep networks.

I suggest the weak acceptance of the paper. After addressing the following remarks, I can adjust my reviews.

1. There are some typos, such as see[?] in page 7, the main theorem on page 6 should be written mathematically with a remark.

2. What is the major technical difference between this paper and Karakida et al., 2018?

3. Here the model is given by conditional probability is defined by a neural network.
The author may also be interested in implicit models, such as normalization flows and generative networks.
In this case, the Wasserstein information matrix, (Hessian of Wasserstein-2 loss), may be very suitable to be studied.
See:

"A. Lin, W.Li, S.Osher, G. Montufar, Wasserstein proximal of GANs, 2018."

"W.Li, G. Montufar, Natural gradient via optimal transport, 2018."



**Experience Assessment:**

I have published one or two papers in this area.

**Review Assessment: Checking Correctness Of Derivations And Theory:**

I carefully checked the derivations and theory.

**Review Assessment: Checking Correctness Of Experiments:**

I assessed the sensibility of the experiments.

**Review Assessment: Thoroughness In Paper Reading:**

I read the paper at least twice and used my best judgement in assessing the paper.

---

> ### Author Response · Authors · 2019-11-15
> **RE reviewer 1**
>
> We thank the reviewer for their careful reading and insightful remarks.
>
> 1. We thank the reviewer for pointing out the typos. Upon reflection, we believe that the theorem would be best presented shortly in the main body, with a longer, more rigorous and easier to follow derivation in the appendix. This will both improve 'flow' of the paper and allow for a more rigorous exposition of our results.
>
> 2. The major technical difference between our submission and Karakida's work [1] is that we do not make any assumptions on the distribution from which the weight matrices are sampled.
>
> Moreover, the intent behind our bound was different from that of Karakida et al. We set out to relate the conditioning of the input-output Jacobian to the Fisher information matrix curvature, while the authors of [1] strove to find an analytical expression for the maximum and mean eigenvalues of the Fisher information matrix under Gaussian weight assumptions. Our results allows us to recover Karakida's Frobenius norm bound (Eq. 16 in [1]) by using traces of random Gaussian matrices to bound the maximal singular of each layer.
>
> Apart from this, we believe that the use of the block Gershgorin theorem might facilitate the analysis of optimization techniques such as K-FAC[2], where a block tri-diagonal approximation of the FIM is used.
>
> 3. Indeed, the Riemannian geometry induced by the $W^2$ distance is highly relevant. We plan to explore it future work, and for the moment we will add reference to Wasserstein information geometry in the background section as alternatives to the Fisher-Rao metric.
>
> [1]R. Karakida, S. Akaho, and S. Amari, “Universal Statistics of Fisher Information in Deep Neural Networks: Mean Field Approach,” in The 22nd International Conference on Artificial Intelligence and Statistics, 2019, pp. 1032–1041.
> [2]J. Martens and R. Grosse, “Optimizing Neural Networks with Kronecker-factored Approximate Curvature,” in International Conference on Machine Learning, 2015, pp. 2408–2417.

---

> > ### Comment · AnonReviewer1 · 2019-11-15
> > **Response to authors**
> >
> > The authors address my questions. I recommend the publication of this paper. I revise the rating from 6 to 8.

---

### Official Review · AnonReviewer2 · 2019-10-24
**Official Blind Review #2**

**Rating:** 6

**Review:**

This paper analyses the training behavior of wide networks and argues orthogonal initialization helps the training. They suggest projections to the manifold of orthogonal weights during training and provide analysis. Their main result seems to be a bound on the eigen-values of the Fisher information matrix for wide networks (Theorem on pg 6). In their experiments they train Stiefel and Oblique networks as examples of manifold constrained networks and claim they converge faster than unconstrained networks.

Cons:
- Page 6, the main theorem of the paper, Theorem (bound on the fisher) doesn’t have a proof.
- Fig 1. What’s the overhead wall-clock time of manifold constraint?, on cifar10 the two manifold don’t have the same rate. Euclidean on cifar10 has higher test accuracy. Test accuracy after 200 epochs is below 90 and below 60.
- There are claims in the paper for providing explanations by making connections to Neural Tangent Kernel but it is mentioned only in the discussion section and they reiterate previously known results.
- Fig 3: is the training plot for cifar10 in this figure the one in figure1? Where is the training curves for svhn? Where should we see the rate of reduction in training loss for these methods?
- Section B.4:  To show that FIM and NTK have the same spectrum you need \nabla^2 L to be identity which is only true for L2 loss function. This does not apply to other loss functions such as cross-entropy.

After rebuttal:
I raise my rating to weak accept. The writing has improved a lot and most of my concerns are addressed. It would be nice if authors could incorporate the timing plots in the appendix.

**Experience Assessment:**

I have read many papers in this area.

**Review Assessment: Checking Correctness Of Derivations And Theory:**

I assessed the sensibility of the derivations and theory.

**Review Assessment: Checking Correctness Of Experiments:**

I assessed the sensibility of the experiments.

**Review Assessment: Thoroughness In Paper Reading:**

I read the paper at least twice and used my best judgement in assessing the paper.

---

> ### Author Response · Authors · 2019-11-15
> **Re:Reviewer 2**
>
> We would like to thank the reviewer for their careful reading of our submission, and their insightful comments.
>
> We would like to stress our main theoretical contribution, which provides a bound on the largest eigenvalues of the Fisher information matrix (FIM).
> This result, generalizes known bound in two ways, it doesn't assume normally distributed weight matrices; it also shows that ensuring that the input-output Jacobian is well conditioned implies that the FIM has a relatively small maximal eigenvalue.

---

> > ### Author Response · Authors · 2019-11-15
> > **re 2**
> >
> > We respectfully disagree in that the proof of the main theorem is missing - it is gradually built up throughout Lemmas 1-3 and the Proposition 1 and Assumption 2.
> > However, we do accede that the proof ought to be more accessible, and to that end we propose to give a short statement in the main body, and relegate most of the details
> > to the appendix, where it will be presented in more detail.
> >
> > Below we present figures showing representative the wall-clock times for examples of each network type. From these example figures, one can see that orthogonally constrained (Stiefel) optimization takes about twice as long as the stochastic optimizer without any manifold constraints on the weight matrices. Nota bene, our implementation heavily relies on a multicore CPU architecture to offload the matrix factorization. From our experiments, the GPU-only implementation is approximately 4 times as slow as the unconstrained descent method. All the numerical experiments were performed on a Tesla P-100 with a Intel(R) Xeon(R) CPU E5-2690 v4 @ 2.60GHz and 128GB RAM.
> >
> > Regarding comparisons to highly performant models, we do not consider the networks analyzed in our submission to be practical, but rather consider them as tools to aide the understanding of training very deep and wide networks. The time complexity of a gradient update for each Stiefel layer is O(m^3) where m is the number of neurons, for the Oblique layers that complexity is O(m^2); which hampers the application of manifold constrained optimization in very wide architectures that have recently been proposed.In our theoretical experiments, we focus on multi-layer perceptrons; and while they do not attain competitive performance with convolutional neural networks, they are far more amenable to theoretical analysis. This has been a dominant approach in works like the Neural Tangent Kernel[3], and the random matrix analyses [1,4,5]. We diligently recreated previously reported performance with similar architecture, viz [1] and [2], with a Bayesian hyperparameter search algorithm.
> >
> > The claims relating the Neural Tangent kernel were intentionally put in the appendix. We do not claim to make novel contributions, but rather report empirical results that can be explained by using recently obtained bounds on the stability of linearizing wide neural networks around their initial parameters. We do however argue that the interpretation of those results is novel, since the focus in literature thus far has been on the stability of the NTK. Here we observe experimentally that networks with a "nearly isotropic" parameter space often train poorly, which goes against intuition from convex optimization. We connect this to the previously known results.
> >
> > The training curves for both CIFAR 10 and SVHN are the left hand panels in Figure 1, and they illustrate that the manifold constrained networks "reduce training loss faster" but do not benefit from an equivalent increase in test accuracy. We apologize for the confusing labelling of the figures, we will clearly state that the left-hand panels are the per sample cross-entropies during training.
> >
> > Lastly, we apologize for the imprecise phrasing -- we agree that the Neural Tangent Kernel only coincides with the FIM for Gaussian likelihoods and l^2 losses. We will make it clearly that this is a qualified equivalence between the two matrices.

---

> > > ### Author Response · Authors · 2019-11-15
> > > **re 3**
> > >
> > >
> > >                      Average training cross-entropy over time
> > >   0.0024 +--------+---------+--------+---------+--------+---------+--------+
> > >          *                                        Stiefel h0 = 9e-4 ****** |
> > >   0.0022 *                                                                 +
> > >          *                                                                 |
> > >    0.002 ***                                                               +
> > >          ****                                                              |
> > >   0.0018 +****                                                             +
> > >          | *****                                                           |
> > >   0.0016 +   *****                                                         +
> > >   0.0014 +     *****                                                       +
> > >          |      ******                                                     |
> > >   0.0012 +        ********                                                 +
> > >          |          **********                                             |
> > >    0.001 +             ************** *                                    +
> > >          |               * *****************                               |
> > >   0.0008 +                     ****************************  * * *         +
> > >          |                          ********************************       |
> > >   0.0006 +                                     *********************       +
> > >          |                                                                 |
> > >   0.0004 +--------+---------+--------+---------+--------+---------+--------+
> > >          0        50       100      150       200      250       300      350
> > >                                     Time [minutes]
> > >
> > >   0.0024 +------------+------------+-------------+------------+------------+
> > >          *                                        Oblique h0 = 9e-4 ****** |
> > >   0.0022 *                                                                 +
> > >          *                                                                 |
> > >    0.002 *                                                                 +
> > >          **                                                                |
> > >   0.0018 ***                                                               +
> > >          |****                                                             |
> > >   0.0016 + ****                                                            +
> > >   0.0014 +   ****                                                          +
> > >          |    *****                                                        |
> > >   0.0012 +     *******                                                     +
> > >          |       *******                                                   |
> > >    0.001 +          ******                                                 +
> > >          |            ********                                             |
> > >   0.0008 +              ***********                                        +
> > >          |                 *******************  *                          |
> > >   0.0006 +                    **************************************       +
> > >          |                           * *****************************       |
> > >   0.0004 +------------+------------+-------------+--****-***********-------+
> > >          0            50          100           150          200          250
> > >                                     Time [minutes]

---

> > > > ### Author Response · Authors · 2019-11-15
> > > > **re4**
> > > >
> > > >
> > > >
> > > > [1]J. Pennington, S. Schoenholz, and S. Ganguli, “Resurrecting the sigmoid in deep learning through dynamical isometry: theory and practice,” in Advances in Neural Information Processing Systems 30, I. Guyon, U. V. Luxburg, S. Bengio, H. Wallach, R. Fergus, S. Vishwanathan, and R. Garnett, Eds. Curran Associates, Inc., 2017, pp. 4785–4795.
> > > > [2]S. Hayou, A. Doucet, and J. Rousseau, “On the Impact of the Activation function on Deep Neural Networks Training,” in International Conference on Machine Learning, 2019, pp. 2672–2680.

---

### Decision · Program_Chairs · 2019-12-19

**Decision:**

Accept (Poster)

**Comment:**

I've gone over this paper carefully and think it's above the bar for ICLR.

The paper proves a relationship between the eigenvalues of the Fisher information matrix and the singular values of the network Jacobian. The main step is bounding the eigenvalues of the full Fisher matrix in terms of the eigenvalues and singular values of individual blocks using Gersgorin disks. The analysis seems correct and (to the best of my knowledge) novel, and relationships between the Jacobian and FIM are interesting insofar as they give different ways of looking at linearized approximations. The Gersgorin disk analysis seems like it may give loose bounds, but the analysis still matches up well with the experiments.

The paper is not quite as strong when it comes to relating the anslysis to optimization. The maximum eigenvalue of the FIM by itself doesn't tell us much about the difficulty of optimization. E.g., if the top FIM eigenvalue is increased, but the distance the weights need to travel is proportionately decreased (as seems plausible when the Jacobian scale is changed), then one could make just as fast progress with a smaller learning rate. So in this light, it's not too surprising that the analysis fails to capture the optimization dynamics once the learning rates are tuned. But despite this limitation, the contribution still seems worthwhile.

The writing can still be improved.

The claim about stability of the linearization explaining the training dynamics appears fairly speculative, and not closely related to the analysis and experiments. I recommend removing it, or at least removing it from the abstract.